# BSSR: BINARIZATION AND SPARSITY FOR IMAGE SUPER-RESOLUTION

## ABSTRACT

Lighter models and faster inference remain the focus in the field of image super-resolution. Quantization and pruning are both effective methods for compressing deep models. Unfortunately, existing approaches often optimize quantization and pruning independently: standalone binarization reduces storage but underutilizes sparsity, while N:M sparsity on weights accelerates inference but leaves high-bit storage overhead. Notably, no prior work has explored N:M sparse binary SR networks. In this paper, we combine quantization and sparsity to propose an extreme compression method for super-resolution tasks, namely BSSR. Within this framework, we introduce two key components: Binarized N:M Sparse Quantizer (BSQ) and Binarized Sparse Gradient Adjuster (BSGA). Firstly, BSQ is a sparse binarization operation across dimensions, simultaneously performing activation and weight binarization while imposing N:M sparsity on weights, significantly reducing storage and computational resource requirements. Secondly, BSGA employs a learnable hyperbolic tangent function combined with distinct gradient scaling factors for preserved and masked elements to address the non-differentiability of binarization and N:M sparse masking, enabling smooth and stable gradient propagation and improving convergence in sparse binary networks. Extensive experiments on SR benchmarks demonstrate that BSSR achieves state-of-the-art performance, outperforming the second-best algorithm by 0.22 dB in PSNR at 4x scaling in MambaIRv2-light compression, and improving PSNR by 0.32 dB at 4x scaling in SwinIR-light compression on the Urban100 dataset.

## 1 INTRODUCTION

Single-image super-resolution (SISR), as a classical low-level computer vision task, has been extensively studied and continuously achieves new state-of-the-art performance with the rapid development of deep neural networks. Early SISR methods primarily relied on convolutional neural networks (CNNs) Lim et al. (2017), which excel at capturing local spatial correlations. With the emergence of Visual Transformers (ViTs), attention-based architectures have become mainstream due to their powerful ability to model long-range dependencies Liang et al. (2021). More recently, selective state space models (SSMs), which efficiently capture long-range dependencies with linear complexity relative to sequence length, have shown remarkable potential as a new backbone for image restoration Guo et al. (2024a; 2025). Despite these advances, the high computational cost and large weight storage of full-precision models remain major obstacles for deploying SISR networks in resource-constrained environments. Consequently, model compression has become an essential step for practical deployment. Existing compression techniques, including quantization Jacob et al. (2018), knowledge distillation Hinton et al. (2015), and pruning Han et al. (2015), aim to reduce storage and computation while preserving model performance. Traditional compression methods such as pruning, low-rank decomposition, or quantization have achieved notable success in classification and detection tasks; however, directly applying these approaches to SISR models often leads to convergence difficulties and performance degradation, primarily due to the pixel-level precision sensitivity and the complex nonlinear feature representation requirements of super-resolution tasks.

Binary quantization, as an extreme compression strategy, can significantly reduce model storage and computational overhead. In classification tasks, methods such as XNOR-Net and Bi-Real Net have successfully binarized both weights and activations. When both weights and activations are quantized to 1 bit (full binary quantization), efficient bitwise operations, such as XNOR and bit

counting, can replace matrix multiplication, enabling maximal acceleration Rastegari et al. (2016). However, directly applying binary quantization to super-resolution models presents significant challenges. SISR activations typically exhibit wider and more continuous distributions and are highly sensitive to minor numerical variations, causing traditional binary methods to suffer from severe information loss and performance degradation. To address these issues, recent studies have proposed SR-specific binarization strategies, including dynamic-threshold quantization and feature-adaptive binarization. Additionally, researchers have explored binary quantization for CNNs Xin et al. (2020; 2023); Xia et al. (2023) and Transformer models Li et al. (2024). Nonetheless, full binary quantization for the Mamba model remains largely unexplored, highlighting the need for further investigation that combines binarization with sparsity to achieve more efficient SR compression.

Pruning represents another effective model compression strategy. Existing works have explored the sparsity in Convolutional Neural Networks (CNNs), as well as the prediction of a pixel-level redundancy mask to improve the inference efficiency of SR networks Wang et al. (2021). Furthermore, pruning methods such as DRC Guo et al. (2024b) involve search-and-prune procedures followed by finetuning, which adds substantial overhead. While the performance of these methods is promising, they fail to fully leverage GPU acceleration and have not been fully adapted to other architectures. N:M structured sparsity offers significant hardware acceleration advantages. For instance, NVIDIA Ampere GPUs can multiply a 2:4 sparse matrix by a dense matrix nearly twice as fast as multiplying two dense matrices. SR-STE Zhou et al. (2021) represents a classical full-precision 2:4 sparse pretraining scheme based on the Straight-Through Estimator (STE), stabilized with additional regularization. S-STE Hu et al. (2024) investigates a 2:4 sparse pretraining scheme under FP8 quantization, but FP8 still incurs substantial GPU memory costs. Existing work primarily focuses on optimizing high-bit sparse weights or exploring structured pruning on medium-sized SR models. However, how to simultaneously achieve efficient sparsity and extreme quantization remains an unexplored challenge. Notably, 1-bit quantized 2:4 sparse pretraining for SR models has yet to be fully explored, highlighting the potential of combining extreme quantization with hardware-efficient sparsity.

Recent studies show that sparsity and 8-bit quantization are non-orthogonal Harma et al. (2024). Applying sparsity before quantization (S→Q) preserves the relative importance of weights and reduces quantization error, which is especially important in super-resolution tasks requiring fine-grained weight adjustments. Since N:M sparsity retains $N$ non-zero weights out of $M$, while standard binarization uses a global scaling factor, a mismatch arises that may harm reconstruction quality. To address this, group-wise quantization computes scaling factors per group, better capturing local weight distributions, preserving N:M sparsity, and maintaining high-fidelity image reconstruction.

In this paper, to address the aforementioned challenges, we propose BSSR: Binarization and Sparsity For Image Super-Resolution. BSSR introduces two key components: Binarized N:M Sparse Quantizer (BSQ) and Binarized Sparse Gradient Adjuster (BSGA). Firstly, BSQ is a sparse binarization operation across dimensions with group-wise scaling factors, simultaneously performing activation and weight binarization while imposing N:M sparsity on weights, significantly reducing storage and computational resource requirements. Secondly, BSGA employs a learnable hyperbolic tangent function combined with distinct gradient scaling factors for preserved and masked elements to address the non-differentiability of binarization and N:M sparse masking, thereby enabling stable gradient propagation and improving training convergence in sparse binary networks. Extensive experiments on SR benchmarks demonstrate that BSSR achieves state-of-the-art performance. Our contributions are summarized as follows:

1. **Binarized N:M Sparse Quantizer (BSQ):** We propose a sparse binarization operation that simultaneously binarizes activations and weights with N:M sparsity, introducing a group-wise adaptive scaling factors, ensuring accurate sparse binary weight approximation, significantly reducing storage and computational resource requirements on edge devices.

2. **Binarized Sparse Gradient Adjuster (BSGA):** We design an adaptive gradient estimator to handle the non-differentiability of binarization and N:M sparse masks. BSGA employs a trainable clipping interval and separate gradient scaling factors for preserved and masked elements, enabling stable and precise gradient propagation and stabilizing training.

3. **State-of-the-art Performance:** Extensive experiments on SR benchmarks demonstrate that BSSR achieves state-of-the-art performance, outperforming the second-best algorithm by 0.22 dB in PSNR at 4x scaling in MambaIRv2-light compression, and improving PSNR by 0.41 dB at 4x scaling in SwinIR-light compression on the Urban100 dataset.

## 2 RELATED WORKS

### 2.1 IMAGE SUPER-RESOLUTION

Image Super-Resolution (SR) aims to reconstruct high-resolution images from low-resolution inputs and has witnessed significant progress with the development of deep learning. Early attempts usually adopt Convolutional Neural Networks (CNNs), such as SRCNN Dong et al. (2014) for image super-resolution, DnCNN Zhang et al. (2017) for image denoising, and ARCNN Dong et al. (2015) for JPEG compression artifact reduction. To further enhance the performance of CNN-based methods, various techniques have been introduced. For instance, EDSR Lim et al. (2017) improves upon residual networks by removing unnecessary modules and expanding network capacity, RDN Zhang et al. (2018b) employs dense connections to enhance representation ability, RCAN Zhang et al. (2018a) introduces channel attention for selecting salient channels, and SAN Dai et al. (2019) leverages second-order attention for performance improvement. Despite the remarkable progress of CNN-based methods, the convolution operator inherently restricts the receptive field to the local kernel, thereby limiting interactions between distant pixels. Recently, Transformer-based architectures have been introduced to SR, leveraging self-attention mechanisms for modeling long-range dependencies. IPT Hu et al. (2021) divides an image into several small patches and processes each patch independently with self-attention. SwinIR Liang et al. (2021) adopts the Swin Transformer backbone and introduces the shifted window self-attention Liu et al. (2021), effectively capturing both local and global contexts and achieving superior performance over CNNs. Uformer Wang et al. (2022) designs a U-shaped Transformer with locally enhanced window attention to better handle image restoration tasks. HAT Chen et al. (2023) incorporates hierarchical attention modules to further improve feature representation. In addition, Restormer Zamir et al. (2022) employs channel-wise self-attention to achieve efficient long-range dependency modeling with reduced complexity. More recently, ATD Guo et al. (2023) employs an adaptive token dictionary to store input-agnostic knowledge, enabling attention to access information beyond the local window. More recently, lightweight and efficient architectures tailored for practical deployment have emerged. Distinct from conventional convolution- or attention-dominated designs, Mamba introduces selective state-space models that enable long-range dependency modeling with linear complexity. The MambaIRv2 Guo et al. (2025) represents hybrid Mamba-Transformer architectures, combining the local feature extraction strengths of Mamba modules with the global modeling capabilities of Transformers. Deploying full-precision hybrid Mamba-Transformer architectures on edge devices incurs high memory, computational, and energy overhead, requiring extreme compression methods for efficiency.

### 2.2 NETWORK BINARIZATION

In recent years, neural-network quantization has emerged as a key approach for efficient deployment on resource-constrained devices, with Binary Neural Networks (BNNs) Hubara et al. (2016) pioneering the quantization of both weights and activations to ±1 using the sign function and the Straight-Through Estimator (STE), achieving substantial reductions in memory footprint and computational cost. Building upon this foundation, XNOR-Net Rastegari et al. (2016) enhanced the representational capacity of 1-bit convolutions by introducing a learnable per-tensor scaling factor, mitigating the information loss inherent in binarization. DoReFa-Net Zhou et al. (2016) further generalized this concept into a unified low-bitwidth quantization framework supporting arbitrary bit-widths for weights, activations, and gradients, systematically analyzing the impact of bit-width on training stability and accuracy. Bi-Real Net Liu et al. (2018) leveraged residual connections and shortcuts to enhance representational power and training stability in binary networks. RTN Li et al. (2020) applied a tunable truncation function to balance accuracy and stability during weight binarization. ReActNet Liu et al. (2020) incorporated RPReLU activations and the RSign operator to mitigate distribution shifts and sign-function information loss, narrowing the accuracy gap with full-precision models. Re-STE Wu et al. (2023) introduced a power-function correction term within STE, enabling flexible trade-offs between error and stability. More recently, BiPer Vargas et al. (2024) employed binary periodic functions for forward propagation while using corresponding sine functions as differentiable proxies in backward propagation, further improving gradient approximation. Collectively, these works illustrate the evolution of network binarization techniques, progressively improving both efficiency and accuracy for practical deployment. Despite the advancements in network binarization, it still has limitations in representational power, necessitating the integration of other compression methods, such as pruning, to further optimize efficiency and performance.

## 2.3 NETWORK PRUNING.

Network pruning aims to remove redundant weights from dense models to reduce computational and storage costs. Auto-Train-Once (ATO) Wu et al. (2024) introduces a controller network that dynamically generates binary masks to guide pruning automatically, eliminating the need for extra fine-tuning. Dual Regression Compression (DRC) Guo et al. (2024b) reduces model redundancy at both layer and channel levels through structured channel pruning. Among pre-training pruning techniques, N:M sparsity, also known as fine-grained structured sparsity, shows great potential, with Nvidia demonstrating a 2× theoretical speedup on Ampere GPUs using 2:4 sparsity for post-training and inference. To accelerate pre-training, Nvidia Mishra et al. (2021) proposed the ASP paradigm, achieving 2:4 sparsity in three steps while conserving training resources. SR-STE Zhou et al. (2021) was the first to train N:M fine-grained sparse networks from scratch by extending the Straight-Through Estimator with a regularization term to mitigate ineffective sparse updates. T-mask Hubara et al. (2021) introduces a transposable sparse mask that accelerates both forward and backward propagation in N:M structured sparse networks, improving training efficiency and model performance. Bi-directional Masks (Bi-Mask) Zhang et al. (2023) separate sparse masks for forward and backward propagation and introduce an efficient weight row permutation to maintain performance while accelerating training. S-STE Hu et al. (2024) continuously projects dense weights to 2:4 sparsity and rescales sparse weights per tensor using a fixed factor for FP8 pre-training. WANDA Sun et al. (2023) leverages weight and activation distribution awareness to guide structured pruning, improving sparsity efficiency while preserving model accuracy. SparseGPT Frantar & Alistarh (2023) performs one-shot structured pruning for large language models by analyzing the Hessian of weights, enabling high sparsity with minimal accuracy loss. Despite significant progress in reducing computational and storage overhead through network pruning, several limitations remain. In particular, weights remain in high-bitwidth representation after pruning alone, failing to effectively reduce storage costs. This motivates the focus of our work: combining pruning with binarization to simultaneously optimize computational efficiency and storage overhead.

## 3 METHOD

This section first presents the definition and training objectives of Binarized N:M Fine-Grained Structured Sparse Networks. We then analyze the limitations of existing binarization and N:M sparsity training methods under extreme compression. Subsequently, we propose a Binarized N:M Sparse method for super-resolution network training, referred to as BSSR. The overview framework of BSSR is shown in the Figure 1. BSSR primarily comprises two core techniques: *Binarized N:M Sparse Quantizer (BSQ)* and *Binarized Sparse Gradient Adjuster (BSGA)*. BSQ simultaneously binarizes activations and imposes N:M sparsity on weights, enabling effective compression while preserving important structural and textural information. BSGA provides adaptive gradient adjustment for the non-differentiable binarization and sparse masks, ensuring stable optimization under extreme compression.

### 3.1 BINARIZED N:M FINE-GRAINED STRUCTURED SPARSE NETWORKS

A binarized N:M sparse network satisfies three simultaneous constraints: N:M sparsity, where in every sliding block of $M$ consecutive weights at most $N$ are non-zero; binary weights, where every non-zero weight is restricted to $\{-1, +1\}$; and binary activations, where activations are binarized to achieve end-to-end binarization. Training a binarized N:M sparse network involves minimizing a loss function over the trainable parameter vector $\mathbf{w} \in \mathbb{R}^d$. The feasible set is defined as:

$$\mathcal{C} = \left\{ \mathbf{w} \in \mathbb{R}^d \mid \forall \text{non-overlapping } M\text{-blocks } \mathcal{B}, \|\mathbf{w}_\mathcal{B}\|_0 = N, \ w_i \in \{-1, +1\} \text{ if } w_i \neq 0 \right\}. \quad (1)$$

The training objective is to minimize the expected loss:

$$\min_{\mathbf{w} \in \mathcal{C}} \ L(\mathbf{w}) = \mathbb{E}_{(\mathbf{x}, y) \sim \mathcal{D}} \left[ \ell(f(\mathbf{x}; \mathbf{w}), y) \right], \quad (2)$$

where $\mathcal{D}$ denotes the training data, $\ell(\cdot, \cdot)$ is the task-specific loss function, and $f(\cdot; \mathbf{w})$ represents the forward function.

## 3.2 BSSR FOR BINARIZED N:M SPARSE IMAGE SUPER-RESOLUTION NETWORK TRAINING

In the SwinIR Liang et al. (2021) and MambaIRv2 Guo et al. (2025) architectures, linear layer parameters are highly dense, constituting the primary computational overhead of the network. For instance, linear layers in SwinIR are distributed across MSA and MLP modules, while MambaIRv2 additionally incorporates ASSM modules. To achieve extreme compression of linear layers, we propose the Binarized N:M Sparse Quantizer (BSQ): it binarizes activation values and applies N:M sparsity while quantizing weights to binary values, significantly reducing storage and computation while preserving model performance. Furthermore, to ensure training stability, we design Binarized Sparse Gradient Adjuster (BSGA) to balance the impact of N:M masking and binarization operations on weight updates. The overall structure of our proposed BSSR method are illustrated in Figure 1.

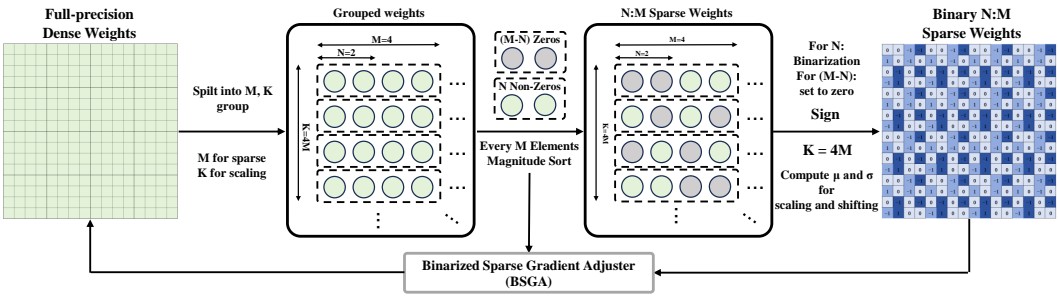

Figure 1: Overview of the proposed **BSSR** framework.

**Binarized N:M Sparse Quantizer (BSQ).** BSQ consists of two parts: *activation binarization* and *weight binarization with N:M sparsity*. Given an activation tensor $X \in \mathbb{R}^{B \times d}$, we first apply a learnable bias module Move$(\cdot)$ and a trainable scaling parameter $\gamma$, then quantize the result into binary values: $X_b = \text{Sign}(\text{Move}(\gamma \cdot X))$, where $\gamma$ is initialized as 1 and updated during training, enabling dynamic adjustment of the quantization boundary. For a weight matrix $W \in \mathbb{R}^{d_{\text{out}} \times d_{\text{in}}}$, we first enforce an N:M sparsity mask, where in every group of $M$ elements only the top-$N$ elements (by magnitude) are preserved: $\mathcal{M}_i = \text{TopNMask}(W_i, N, M)$. To address the granularity mismatch between N:M sparsity and conventional quantization, we introduce the *Group-wise Sparse Binarizer (GSB)*. Instead of using a single global scaling factor for the entire weight tensor, the Group-wise Sparse Binarizer (GSB) divides the weights into $K$ groups, where $K$ is an integer multiple of $M$, and computes a separate scaling factor for each group:

$$s_k = \frac{1}{N} \sum_{j=1}^{M} |W_{k,j} \cdot \mathcal{M}_{k,j}|, \quad k = 1, \dots, K, \tag{3}$$

where $\mathcal{M}_{k,j} \in \{0, 1\}$ represents the N:M sparsity mask.

Within each group, the surviving (non-zero) weights are first mean-centered based on the group mean and then binarized using the group-specific scaling factor:

$$\hat{W}_{k,j} = s_k \cdot \text{Sign}\Big((W_{k,j} - \mu_k) \cdot \mathcal{M}_{k,j}\Big), \tag{4}$$

where the group mean $\mu_k$ is computed only over the preserved weights in group $k$:

$$\mu_k = \frac{1}{N} \sum_{j=1}^{M} W_{k,j} \cdot \mathcal{M}_{k,j}. \tag{5}$$

This group-wise normalization and scaling ensures that each group is independently adapted to its local distribution, reducing quantization error and improving sparse binary weights approximation.

**Binarized Sparse Gradient Adjuster (BSGA).** Due to the inherent discontinuity of the binarization operation and the zero-filling of N:M sparse masks, standard derivatives cannot be directly applied during backpropagation. Specifically, the derivative of the `sign` function is an impulse function,

making it non-differentiable. In practice, gradients are typically approximated using either a clipped function or a $\texttt{tanh}$-based approximation. BSGA employs a learnable hyperbolic tangent function to approximate the gradient:

$$E(X) = \tanh(\kappa X), \qquad E'(X) = \kappa\big(1 - \tanh^2(\kappa X)\big), \tag{6}$$

where $\kappa$ is a learnable parameter controlling the slope of the approximation. During backpropagation, the surrogate gradient with respect to the latent weight $\tilde{W}_t$ is given by

$$\frac{\partial \mathcal{L}}{\partial \tilde{W}_t} \approx \frac{\partial \mathcal{L}}{\partial \hat{W}_t} \odot E'(\tilde{W}_t) = G_t \odot \Big(\kappa\big(1 - \tanh^2(\kappa \tilde{W}_t)\big)\Big), \tag{7}$$

with $G_t = \frac{\partial \mathcal{L}}{\partial \hat{W}_t}$. Moreover, to explicitly distinguish the preserved (non-zero) elements from the masked (zeroed) ones within each N:M group, we introduce separate gradient scaling strategies. Preserved weights are updated using the $\texttt{tanh}$-based surrogate gradient, while masked weights are softly regularized toward zero. This leads to the following update rule:

$$\tilde{W}_{t+1} = \tilde{W}_t - \gamma_t\Big(\mathcal{M} \odot \big(G_t \odot E'(\tilde{W}_t)\big) + \rho\big((1 - \mathcal{M}) \odot \tilde{W}_t\big)\Big), \tag{8}$$

where $\mathcal{M}$ is the N:M sparse mask, $\gamma_t$ is the learning rate, and $\rho$ is the regularization coefficient. This formulation provides smooth and stable gradients for both binary and masked elements while preserving the N:M sparsity structure. This allows the network to maintain meaningful updates for the non-zero weights while preventing gradient updates on the pruned weights, ensuring stable training. BSGA jointly considers the characteristics of binarization and N:M sparsity by integrating adaptive clipping and element-wise gradient scaling. By dynamically adjusting the gradient range for each element type and preserving the structure of sparse groups, BSGA achieves more accurate and stable gradient propagation. This leads to improved convergence behavior when training sparse binary networks and reduces the risk of gradient explosion or vanishing.

**Overall Method.** By jointly applying the BSQ and BSGA, our BSSR framework establishes a new paradigm for training binarized N:M sparse networks. BSSR simultaneously achieves storage and computation compression, stabilizes training, and preserves state-of-the-art super-resolution performance of binarized N:M sparse networks, making it a practical and efficient solution for deploying SR models on resource-constrained edge devices.

## 4 EXPERIMENTS

In this section, we demonstrate the effectiveness of the proposed BSSR training scheme on the SISR task. Swin-IR-light Liang et al. (2021) is selected as the representative transformer model, while MambaIRv2-light Guo et al. (2025) serves as the representative mamba model.

### 4.1 EXPERIMENTAL SETTINGS

For quantitative comparison, we evaluated various super-resolution (SR) methods on five benchmark datasets: Set5 Bevilacqua et al. (2012), Set14 Zeyde et al. (2010), B100 Martin et al. (2001), Urban100 Huang et al. (2015), and Manga109 Matsui et al. (2017). We employ two commonly used metrics: PSNR and SSIM Wang et al. (2004), as standards for measuring the quality of super-resolved images, calculated on the luminance (Y) component in the YCbCr color space. The data augmentation technique employed horizontal flipping along with random rotations of $90°$, $180°$, and $270°$. Training data was obtained from $64 \times 64$ RGB input blocks of LR images and their corresponding HR blocks. The model was trained for 100k iterations using the Adam optimizer Kingma (2014), with parameters $\beta_1 = 0.9$ and $\beta_2 = 0.99$, and the batch-size set to 16. The learning rate was initially set to $2 \times 10^{-4}$ and halved at specified iteration milestones using a cosine annealing strategy. All experiments were conducted with identical parameter settings to ensure fair comparison. This work is implemented based on the PaddlePaddle framework, and experiments are conducted on an NVIDIA RTX 4090 GPU.

### 4.2 COMPARISON RESULTS

We first deployed the proposed BSSR method in the Swin-IR-light and MambaIRv2-light models, where the weights of the linear layers underwent 2:4 sparse 1-bit quantization, and all activations

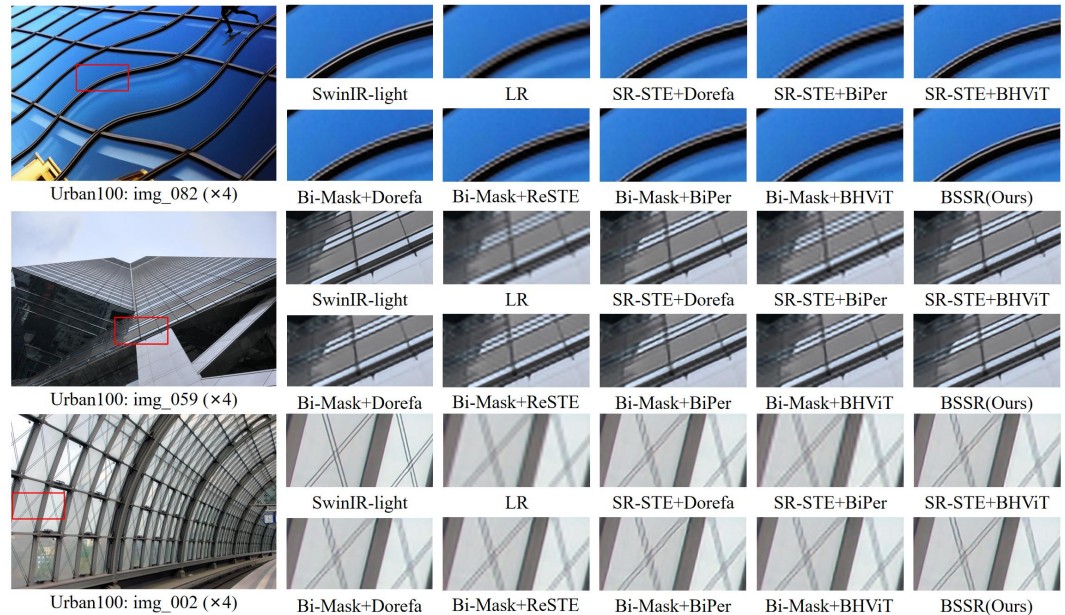

Figure 2: Visual comparison (×4). We compare our BSSR with recent combined quantization and pruning methods. The results show that BSSR performs clearly better than other methods in all cases.

were quantized to 1 bit. For comparison, we considered several representative binary quantization methods, including DoReFa Zhou et al. (2016), Re-STE Wu et al. (2023), BiPer Vargas et al. (2024), and BHViT Gao et al. (2025). In addition, the 2:4 sparsification methods selected were SR-STE Zhou et al. (2021) and Bi-Mask Zhang et al. (2023), while for channel pruning in linear layers, we adopted the DRC method Guo et al. (2024b). These methods cover a wide range of existing model compression strategies, allowing for a comprehensive assessment of our approach. Table 1 presents a comprehensive comparison of various combination of binary quantization and pruning methods together with our approach, covering super-resolution scales of ×2 and ×4. The ×4 visual comparison results are shown in Figure 2.

**Qualitative results.** Figure 2 presents the visual comparison results for ×4 super-resolution. Overall, the results demonstrate that BSSR can effectively model complex activation distributions, preserve structural details and object boundaries, and significantly reduce blurring and distortion. The exceptional performance of BSSR stems from its two core modules: the Binarized N:M Sparse Quantizer (BSQ) and the Binarized Sparse Gradient Adjuster (BSGA). BSQ ensures efficient compression of activation values and weights while preserving critical texture and structural information. Meanwhile, BSGA effectively mitigates optimization challenges posed by non-differentiable binarization and sparse masks through adaptive gradient adjustment. These modules synergistically enhance not only standard quantization metrics like PSNR and SSIM but also significantly improve the quality of reconstructed images. For example, on *img_082*, *img_059*, and *img_002* from the Urban100 dataset, our method generates visual results most similar to the full-precision model, whereas competing approaches suffer from varying degrees of edge diffusion and texture loss, leading to noticeably lower image fidelity. These observations confirm that BSSR effectively balances aggressive compression with high-quality visual reconstruction, making it a robust solution for extreme-compression super-resolution models.

**Quantitative results.** Our BSSR achieved the optimal PSNR/SSIM scores compared to the comparison methods in the mambairv2-light model compression experiments at scales ×2 and ×4 across all five benchmark datasets, demonstrating superior reconstruction accuracy and visual fidelity compared to the other methods. Compared to other methods combining binary quantization with DRC, BSSR eliminates the cumbersome process of channel search → channel pruning → fine-tuning while achieving superior results. Compared to existing N:M sparse fusion methods with binary

| Method | Scale | Set5 | | Set14 | | B100 | | Urban100 | | Manga109 | |
|---|---|---|---|---|---|---|---|---|---|---|---|
| | | PSNR | SSIM | PSNR | SSIM | PSNR | SSIM | PSNR | SSIM | PSNR | SSIM |
| MambaIRv2-light | x2 | 38.24 | 0.9615 | 34.05 | 0.9222 | 32.33 | 0.9017 | 34.11 | 0.9452 | 39.34 | 0.9785 |
| DoReFa + SR-STE | x2 | 37.23 | 0.9573 | 32.87 | 0.9107 | 31.61 | 0.8916 | 31.29 | 0.9194 | 36.90 | 0.9724 |
| Re-STE + SR-STE | x2 | 37.37 | 0.9582 | 33.02 | 0.9122 | 31.73 | 0.8936 | 31.60 | 0.9233 | 37.00 | 0.9731 |
| BiPer + SR-STE | x2 | 36.60 | 0.9545 | 32.34 | 0.9059 | 31.20 | 0.8861 | 30.02 | 0.9035 | 35.16 | 0.9656 |
| BHViT + SR-STE | x2 | 37.30 | 0.9579 | 32.99 | 0.9125 | 31.71 | 0.8936 | 31.57 | 0.9231 | 36.82 | 0.9726 |
| DoReFa + Bi-Mask | x2 | 37.16 | 0.9572 | 32.87 | 0.9109 | 31.63 | 0.8923 | 31.33 | 0.9198 | 36.83 | 0.9724 |
| Re-STE + Bi-Mask | x2 | 37.18 | 0.9571 | 32.79 | 0.9102 | 31.55 | 0.8912 | 31.00 | 0.9163 | 36.65 | 0.9713 |
| BiPer + Bi-Mask | x2 | 37.15 | 0.9570 | 32.83 | 0.9105 | 31.61 | 0.8919 | 31.13 | 0.9178 | 36.49 | 0.9711 |
| BHViT + Bi-Mask | x2 | 37.34 | 0.9581 | 33.03 | 0.9127 | 31.72 | 0.8937 | 31.67 | 0.9235 | 37.16 | 0.9734 |
| DoReFa + DRC | x2 | 37.15 | 0.9552 | 32.84 | 0.9092 | 31.59 | 0.8892 | 31.24 | 0.9176 | 36.82 | 0.9699 |
| Re-STE + DRC | x2 | 37.16 | 0.9561 | 32.94 | 0.9116 | 31.67 | 0.8914 | 31.55 | 0.9225 | 36.85 | 0.9712 |
| BiPer + DRC | x2 | 37.25 | 0.9562 | 32.90 | 0.9100 | 31.62 | 0.8899 | 31.19 | 0.9174 | 36.66 | 0.9702 |
| BHViT + DRC | x2 | 37.06 | 0.9566 | 32.90 | 0.9118 | 31.67 | 0.8916 | 31.53 | 0.9221 | 36.71 | 0.9716 |
| **BSSR(Ours)** | x2 | **37.56** | **0.9591** | **33.16** | **0.9146** | **31.86** | **0.8955** | **32.03** | **0.9277** | **37.65** | **0.9747** |
| MambaIRv2-light | x4 | 32.51 | 0.8992 | 28.84 | 0.7878 | 27.75 | 0.7426 | 26.82 | 0.8079 | 31.24 | 0.9182 |
| DoReFa + SR-STE | x4 | 30.96 | 0.8757 | 27.82 | 0.7626 | 27.06 | 0.7178 | 25.42 | 0.7567 | 28.46 | 0.8760 |
| Re-STE + SR-STE | x4 | 31.34 | 0.8834 | 28.05 | 0.7688 | 27.22 | 0.7243 | 25.75 | 0.7709 | 28.97 | 0.8872 |
| BiPer + SR-STE | x4 | 30.26 | 0.8611 | 27.31 | 0.7511 | 26.79 | 0.7101 | 24.75 | 0.7317 | 26.99 | 0.8466 |
| BHViT + SR-STE | x4 | 31.22 | 0.8813 | 27.96 | 0.7674 | 27.19 | 0.7234 | 25.69 | 0.7685 | 28.84 | 0.8849 |
| DoReFa + Bi-Mask | x4 | 31.05 | 0.8778 | 27.91 | 0.7648 | 27.11 | 0.7196 | 25.49 | 0.7595 | 28.61 | 0.8794 |
| Re-STE + Bi-Mask | x4 | 30.93 | 0.8759 | 27.77 | 0.7629 | 27.06 | 0.7192 | 25.34 | 0.7552 | 28.24 | 0.8743 |
| BiPer + Bi-Mask | x4 | 31.07 | 0.8781 | 27.87 | 0.7646 | 27.14 | 0.7207 | 25.51 | 0.7614 | 28.53 | 0.8778 |
| BHViT + Bi-Mask | x4 | 31.27 | 0.8824 | 28.04 | 0.7689 | 27.21 | 0.7239 | 25.71 | 0.7686 | 28.90 | 0.8859 |
| DoReFa + DRC | x4 | 30.58 | 0.8640 | 27.56 | 0.7525 | 26.87 | 0.7069 | 25.08 | 0.7414 | 27.56 | 0.8529 |
| Re-STE + DRC | x4 | 31.41 | 0.8814 | 28.06 | 0.7661 | 27.20 | 0.7188 | 25.77 | 0.7681 | 28.96 | 0.8833 |
| BiPer + DRC | x4 | 31.10 | 0.8769 | 27.92 | 0.7628 | 27.09 | 0.7158 | 25.50 | 0.7589 | 28.46 | 0.8748 |
| BHViT + DRC | x4 | 31.32 | 0.8816 | 28.07 | 0.7671 | 27.21 | 0.7196 | 25.77 | 0.7681 | 28.93 | 0.8837 |
| **BSSR(Ours)** | x4 | **31.51** | **0.8858** | **28.18** | **0.7716** | **27.31** | **0.7269** | **25.99** | **0.7773** | **29.27** | **0.8913** |
| Method | Scale | Set5 | | Set14 | | B100 | | Urban100 | | Manga109 | |
| | | PSNR | SSIM | PSNR | SSIM | PSNR | SSIM | PSNR | SSIM | PSNR | SSIM |
| SwinIR-light | x2 | 38.15 | 0.9611 | 33.86 | 0.9206 | 32.31 | 0.9012 | 32.76 | 0.934 | 39.11 | 0.9781 |
| DoReFa + SR-STE | x2 | 37.10 | 0.9569 | 32.76 | 0.9101 | 31.52 | 0.8910 | 30.92 | 0.9151 | 36.34 | 0.9708 |
| Re-STE + SR-STE | x2 | 37.14 | 0.9573 | 32.77 | 0.9109 | 31.56 | 0.8922 | 30.96 | 0.9165 | 36.50 | 0.9716 |
| BiPer + SR-STE | x2 | 35.35 | 0.9462 | 31.61 | 0.8982 | 30.59 | 0.8774 | 28.89 | 0.8860 | 33.15 | 0.9539 |
| BHViT + SR-STE | x2 | 37.13 | 0.9571 | 32.80 | 0.9106 | 31.58 | 0.8920 | 31.06 | 0.9172 | 36.60 | 0.9716 |
| DoReFa + Bi-Mask | x2 | 37.19 | 0.9572 | 32.77 | 0.9102 | 31.53 | 0.8910 | 30.94 | 0.9151 | 36.47 | 0.9711 |
| Re-STE + Bi-Mask | x2 | 36.84 | 0.9555 | 32.55 | 0.9075 | 31.29 | 0.8874 | 30.32 | 0.9060 | 35.90 | 0.9683 |
| BiPer + Bi-Mask | x2 | 36.81 | 0.9554 | 32.60 | 0.9087 | 31.45 | 0.8897 | 30.71 | 0.9128 | 35.98 | 0.9690 |
| BHViT + Bi-Mask | x2 | 37.03 | 0.9567 | 32.72 | 0.9097 | 31.49 | 0.8903 | 30.71 | 0.9132 | 36.31 | 0.9701 |
| DoReFa + DRC | x2 | 37.05 | 0.9555 | 32.73 | 0.9088 | 31.50 | 0.8890 | 30.93 | 0.9150 | 36.34 | 0.9695 |
| BiPer + DRC | x2 | 36.36 | 0.9527 | 32.27 | 0.9054 | 31.11 | 0.8842 | 29.77 | 0.8997 | 34.87 | 0.9632 |
| BHViT + DRC | x2 | 36.96 | 0.9551 | 32.67 | 0.9083 | 31.43 | 0.8874 | 30.69 | 0.9112 | 36.33 | 0.9687 |
| **BSSR(Ours)** | x2 | **37.48** | **0.9585** | **33.06** | **0.9134** | **31.79** | **0.8946** | **31.65** | **0.9237** | **37.31** | **0.9738** |
| SwinIR-light | x4 | 32.45 | 0.8976 | 28.77 | 0.7858 | 27.69 | 0.7406 | 26.48 | 0.798 | 30.92 | 0.915 |
| DoReFa + SR-STE | x4 | 30.88 | 0.8752 | 27.74 | 0.7619 | 27.04 | 0.7177 | 25.28 | 0.7521 | 28.12 | 0.8714 |
| Re-STE + SR-STE | x4 | 31.01 | 0.8770 | 27.79 | 0.7634 | 27.08 | 0.7192 | 25.34 | 0.7555 | 28.27 | 0.8745 |
| BiPer + SR-STE | x4 | 29.34 | 0.8332 | 26.67 | 0.7312 | 26.41 | 0.6948 | 24.12 | 0.7013 | 25.76 | 0.8085 |
| BHViT + SR-STE | x4 | 31.03 | 0.8777 | 27.81 | 0.7639 | 27.09 | 0.7199 | 25.38 | 0.7568 | 28.33 | 0.8763 |
| DoReFa + Bi-Mask | x4 | 30.87 | 0.8746 | 27.75 | 0.7619 | 27.02 | 0.7173 | 25.28 | 0.7524 | 28.17 | 0.8723 |
| Re-STE + Bi-Mask | x4 | 30.45 | 0.8652 | 27.46 | 0.7536 | 26.86 | 0.7111 | 24.91 | 0.7350 | 27.42 | 0.8546 |
| BiPer + Bi-Mask | x4 | 30.63 | 0.8690 | 27.56 | 0.7573 | 26.95 | 0.7145 | 25.11 | 0.7462 | 27.65 | 0.8609 |
| BHViT + Bi-Mask | x4 | 30.70 | 0.8716 | 27.67 | 0.7601 | 27.99 | 0.7167 | 25.18 | 0.7484 | 28.02 | 0.8693 |
| DoReFa + DRC | x4 | 30.84 | 0.8711 | 27.76 | 0.7592 | 27.00 | 0.7124 | 25.26 | 0.7493 | 28.02 | 0.8659 |
| BiPer + DRC | x4 | 30.00 | 0.8499 | 27.13 | 0.7422 | 26.65 | 0.6995 | 24.54 | 0.7200 | 26.51 | 0.8296 |
| BHViT + DRC | x4 | 30.62 | 0.8658 | 27.59 | 0.7542 | 26.91 | 0.7087 | 25.00 | 0.7393 | 27.45 | 0.8538 |
| **BSSR(Ours)** | x4 | **31.27** | **0.8822** | **27.99** | **0.7686** | **27.20** | **0.7237** | **25.70** | **0.7690** | **28.86** | **0.8859** |

Table 1: Quantitative comparison with other methods.

quantization, the BSQ and BSGA modules in BSSR effectively handle adaptive compression in super-resolution models. BSSR exhibits notable improvements on Urban100 and Manga109, indicating its effectiveness in handling scenes with complex textures and rich details. Specifically, in the MambaIRv2-light model compression experiments, our BSSR achieves: an improvement of 0.36 dB / 0.0042 over the second-best algorithm on the Urban100 test set (2x scale), and an improvement of 0.22 dB / 0.0064 over the second-best algorithm on the Urban100 test set (4x scale). In the SwinIR-light model compression experiments on Urban100, our BSSR algorithm achieved optimal PSNR/SSIM scores, improving PSNR/SSIM by 0.59 dB and 0.0065 at 2× scaling, and by 0.32 dB and 0.0122 at 4× scaling, respectively.

| Group size $K$ | Scale | Set5 | | Set14 | | B100 | | Urban100 | | Manga109 | |
|---|---|---|---|---|---|---|---|---|---|---|---|
| | | PSNR | SSIM | PSNR | SSIM | PSNR | SSIM | PSNR | SSIM | PSNR | SSIM |
| $4M$ | ×4 | 31.52 | 0.8859 | 28.18 | 0.7715 | 27.30 | 0.7270 | 26.00 | 0.7773 | 29.28 | 0.8919 |
| $8M$ | ×4 | 31.51 | 0.8858 | 28.18 | 0.7716 | 27.31 | 0.7269 | 25.99 | 0.7773 | 29.27 | 0.8913 |
| $16M$ | ×4 | 31.48 | 0.8830 | 28.15 | 0.7690 | 27.29 | 0.7245 | 25.96 | 0.7755 | 29.24 | 0.8890 |

Table 2: Ablation study on BSQ group size $K$, with results in PSNR (dB) and SSIM.

| $\kappa$ | Scale | Set5 | | Set14 | | B100 | | Urban100 | | Manga109 | |
|---|---|---|---|---|---|---|---|---|---|---|---|
| | | PSNR | SSIM | PSNR | SSIM | PSNR | SSIM | PSNR | SSIM | PSNR | SSIM |
| 0.25 | ×4 | 31.46 | 0.8857 | 28.14 | 0.7710 | 27.29 | 0.7267 | 25.95 | 0.7769 | 29.22 | 0.8908 |
| 0.50 | ×4 | 31.51 | 0.8858 | 28.18 | 0.7716 | 27.31 | 0.7269 | 25.99 | 0.7773 | 29.27 | 0.8913 |
| 2 | ×4 | 31.49 | 0.8840 | 28.16 | 0.7690 | 27.30 | 0.7245 | 25.97 | 0.7765 | 29.25 | 0.8895 |
| 4 | ×4 | 31.44 | 0.8830 | 28.12 | 0.7687 | 27.27 | 0.7242 | 25.92 | 0.7761 | 29.21 | 0.8890 |

| $\rho$ | Scale | Set5 | | Set14 | | B100 | | Urban100 | | Manga109 | |
|---|---|---|---|---|---|---|---|---|---|---|---|
| | | PSNR | SSIM | PSNR | SSIM | PSNR | SSIM | PSNR | SSIM | PSNR | SSIM |
| 0 | ×4 | 31.47 | 0.8857 | 28.13 | 0.7714 | 27.28 | 0.7265 | 25.94 | 0.7771 | 29.24 | 0.8912 |
| 0.002 | ×4 | 31.51 | 0.8858 | 28.18 | 0.7716 | 27.31 | 0.7269 | 25.99 | 0.7773 | 29.27 | 0.8913 |
| 0.005 | ×4 | 31.49 | 0.8847 | 28.16 | 0.7698 | 27.30 | 0.7261 | 25.97 | 0.7767 | 29.25 | 0.8890 |
| -0.002 | ×4 | 31.44 | 0.8851 | 28.15 | 0.7712 | 27.27 | 0.7267 | 25.92 | 0.7768 | 29.20 | 0.8898 |

Table 3: Ablation study on BSGA hyperparameters $\kappa$ and $\rho$, with results in PSNR (dB) and SSIM.

## 5 ABLATION STUDY

To validate the effectiveness of each component in the proposed BSSR framework, we further conduct ablation studies on the proposed Binarized N:M Sparse Quantizer (BSQ) and Binarized Sparse Gradient Adjuster (BSGA). All experiments are conducted on the ×4 super-resolution benchmark, trained for 100K iterations with the same training settings to ensure fair comparison.

**Ablation on BSQ Group Size.** We investigate the impact of group size $K$ in the group-wise sparse binarization scheme on the approximation accuracy of sparse binary weights. In addition to $K = 4M$, $8M$, and $16M$, allowing us to analyze the effect of granularity from fine to coarse. Based on the ablation results in Table 2, we select a group size of $8M$ for BSSR. While $4M$ achieves similar performance, $8M$ slightly improves PSNR on key datasets, and $16M$ shows minor degradation in overall performance. Therefore, $8M$ provides a good trade-off between sparsity and accuracy.

**Ablation Study on BSGA.** We conduct an ablation study to analyze the impact of two key hyperparameters in BSGA: the slope parameter $\kappa$ of the surrogate gradient and the regularization coefficient $\rho$. Based on the ablation study results in Table 3, we select $\kappa = 0.50$ and $\rho = 0.002$ for BSGA, as a moderate $\kappa$ ensures a stable tanh-based approximation of the non-differentiable sign function, preserving binary behavior while maintaining smooth gradient propagation, and a moderate $\rho$ provides effective regularization of masked weights, allowing preserved weights to update properly.

## 6 DISCUSSION AND CONCLUSION

In this work, we proposed BSSR, a unified binarized N:M sparse training framework for image super-resolution. Within BSSR, we introduced two key components: BSQ and BSGA. BSQ performs group-wise sparse binarization of both activations and weights, effectively reducing memory footprint and computational cost while preserving accurate weight approximation. BSGA, on the other hand, tackles the non-differentiability of binarization and N:M sparse masking by employing a learnable hyperbolic tangent function along with separate gradient scaling factors for preserved and masked elements, ensuring stable and smooth gradient propagation during training. Extensive experiments on super-resolution benchmarks demonstrate that BSSR achieves state-of-the-art performance while significantly reducing computational and storage costs, making it highly suitable for deployment of high-performance super-resolution models on resource-constrained devices. In summary, BSSR offers an effective framework for combining sparsity and binarization, paving the way for future research on extreme model compression without sacrificing performance.

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
