# OpenReview forum: "BSSR: Binarization and Sparsity For Image Super-Resolution"
_ICLR.cc/2026/Conference — ICLR 2026 Conference Desk Rejected Submission_

### Official Review · Reviewer_Fh3v · 2025-10-25

**Soundness:** 3
**Presentation:** 3
**Contribution:** 3
**Rating:** 6
**Confidence:** 5

**Summary:**

This paper presents BSSR, a binarized and sparse model lighting method for SISR. The authors propose two components: (1) BSQ for binarizing and sparsing, (2) BSGA for gradient transfer. Experiments are conducted on SwinIR-light and MambaIRv2-light, and commonly used benchmarks. The results are SOTA compared with the combinations of binary quantization methods and sparsification methods.

**Strengths:**

- The writing is clear and easy to follow.
- The increase of PSNR and SSIM on five benchmarks is notable.

**Weaknesses:**

- The experiments are all conducted on the light version of MambaIRv2 and SwinIR. Please provide the results on a bigger version, such as MambaIRv2_SR. Different versions share the same structure. These experiments can be easily conducted and make BSSR more convincing.
-  The core motivation of BSSR is to reduce memory footprint and computation cost. However, the related results are not reported in the paper. Please provide the complexity of the BSSR, including (1) the required GPU memory during training and inference, (2) the number of parameters of the BSSR, and (3) the speedup ratio of BSSR compared with the MambaIRv2-light and SwinIR-light.
- Please analyze the storage and computation overhead of Eq (3,4,5) during training and inference in detail.
- The authors claim that BSGA achieves stable gradient propagation in lines 291~292. Please provide related results to demonstrate "stable".

**Questions:**

- The citing format is incorrect. Please use citep instead of cite.
- What's the relationship between Eq (8) and the Adam optimizer? Adam does not update the parameters with Eq (8).

---

### Official Review · Reviewer_qFGN · 2025-10-27

**Soundness:** 4
**Presentation:** 2
**Contribution:** 2
**Rating:** 2
**Confidence:** 4

**Summary:**

This paper proposes BSSR, a framework combining binarization and N:M sparsity for image super-resolution models. The method consists of two components: (1) the Binarized N:M Sparse Quantizer (BSQ), which applies activation and weight binarization along with group-wise N:M sparsity, and (2) the Binarized Sparse Gradient Adjuster (BSGA), which employs a learnable tanh surrogate gradient with different scaling for preserved and masked weights. The approach is evaluated on recent SR models (SwinIR-light and MambaIRv2-light), showing state-of-the-art performance, achieving consistent PSNR/SSIM gains compared to prior quantization and pruning baselines.

The paper tackles an important direction in extreme compression for SR, and the empirical results are strong. However, the lack of novelty, the absence of hardware-oriented benchmarks, and the limited scope to SR reduce the overall impact.

**Strengths:**

1. The paper evaluates the proposed method on recent lightweight and competitive SR models such as SwinIR-light and MambaIRv2-light, showing strong reconstruction performance improvements.

2. The integration of binarization with N:M sparsity is timely and relevant, addressing efficiency concerns in super-resolution networks.

**Weaknesses:**

1. The method is not SR-specific and could be applied to other tasks. Demonstrating generalizability (e.g., classification, detection) would strengthen the contribution.

2. The motivation of N:M sparsity and binarization lies in hardware efficiency, yet no measurements of memory access, computation cost, or latency are reported. This missing evaluation is critical for assessing the practical value.

3. The paper’s presentation is difficult to read due to overly long sentences and dense writing.

4. N:M sparsity is supported only on Ampere GPUs, limiting deployment applicability.

5. The main components (group-wise sparsity, tanh-based STE) are well-established techniques; combining them provides incremental novelty.

**Questions:**

Please see Weakness and answers the concerns.

**Details Of Ethics Concerns:**

No ethics concerns exisit

---

### Official Review · Reviewer_q78U · 2025-10-27

**Soundness:** 3
**Presentation:** 3
**Contribution:** 2
**Rating:** 4
**Confidence:** 4

**Summary:**

In this paper, the authors propose BSSR by combining binarization and N:M structured sparsity. Specifically, a Binarized N:M Sparse Quantizer (BSQ) is developed to simultaneously perform activation and weight binarization while imposing N:M sparsity on weights. Meanwhile, a Binarized Sparse Gradient Adjuster (BSGA) is introduced to employ learnable hyperbolic tangent function for end-to-end optimization. Experiments are conducted on widely-applied becnhmarks and the results demonstrate that BSSR achieves state-of-the-art performance.

**Strengths:**

- Extensive experiments are conducted and the proposed method produces promising results.
- This paper is well written and easy to follow.

**Weaknesses:**

- Despite promising results, the technical novelty of the proposed method is limited. Network quantization, network binarization, and fine-grained pruning have been widely investigated for efficient image SR. In addition, the combination of these techniques has also been studied. From this perspective, the technical contribution is rather limited.

- Several representation baselines ([c1-c2]) are missing in the performance evaluation. In seems these methods produces even superior performance (both quantitatively and visually) as compared to the proposed method. From this point of view, the effectiveness and superiority of the proposed method should be further validated.

[c1] Lightweight Image Super-Resolution via Flexible Meta Pruning
[c2] Flexible Residual Binarization for Image Super-Resolution

- While this paper focus on efficient image SR, quantitative results on FLOPs, memory cost, and runtime are necessary to demonstrate the effectiveness of the proposed method.

- Ablation experiments are not sufficient. As binarization and structured sparsity are combined in the proposed method, ablation experiments should be conducted to evaluate the accuracy and efficiency performance with only binarization or sparsity.

**Questions:**

See Weaknesses

---

### Official Review · Reviewer_Hd5x · 2025-10-31

**Soundness:** 3
**Presentation:** 4
**Contribution:** 2
**Rating:** 6
**Confidence:** 5

**Summary:**

This paper proposes an extreme compression framework, BSSR, for lightweight image super-resolution (SR), integrating sparsification and quantization method. BSSR comprises a binaried N:M sparse quantizer and a binarized sparse gradient adjustor, which together enhance the trainability of binarized-quantized networks. Experimental results demonstrate the effectiveness of the proposed strategy.

**Strengths:**

1. This paper discusses the effective combination of two different compression strategies, which is a new perspective to solve the problem of lightweight SISR.
2. The manuscript is clearly written with a well-organized structure. The proposed method is concise and effective, demonstrating practical applicability.

**Weaknesses:**

1. Both grouped quantization and weight soft regularization are not new techniques. The authors need to highlight their contributions.

2. The rationale for adopting grouped quantization remains unclear. As shown in Table 2, performance degrades with increasing group numbers, making it difficult to justify the use of grouped quantization.

3. As the authors claim, "BSSR eliminates the cumbersome process of channel search → channel pruning → fine-tuning while achieving superior results." Therefore, I am curious whether the proposed method can reduce training costs compared to single binarization or quantization methods.

4. The experimental results lack comparisons with single methods, making it difficult to evaluate the trade-offs between performance and model size in the combined approach.

**Questions:**

please refer to Weakness

---

### Note · Program_Chairs · 2026-01-17
**Submission Desk Rejected by Program Chairs**

The following references in this submission do not refer to real documents and/or have major errors in bibliographic information:

 Yong Guo, Yulun Wang, Tianyu Qiao, Kai Zhang, Kai Han, Chang Xu, and Dacheng Tao. Attend to dictionary: Transformer learns sparse codes for image restoration. In Advances in Neural Information Processing Systems (NeurIPS), 2023.